# Glucose-Lowering Therapy beyond Insulin in Type 1 Diabetes: A Narrative Review on Existing Evidence from Randomized Controlled Trials and Clinical Perspective

**DOI:** 10.3390/pharmaceutics14061180

**Published:** 2022-05-31

**Authors:** Felix Aberer, Thomas R. Pieber, Max L. Eckstein, Harald Sourij, Othmar Moser

**Affiliations:** 1Division of Endocrinology and Diabetology, Department of Internal Medicine, Medical University of Graz, 8036 Graz, Austria; felix.aberer@medunigraz.at (F.A.); thomas.pieber@medunigraz.at (T.R.P.); othmar.moser@uni-bayreuth.de (O.M.); 2Division of Exercise Physiology and Metabolism, Institute of Sport Science, University of Bayreuth, 95447 Bayreuth, Germany; max.eckstein@uni-bayreuth.de

**Keywords:** Type 1 diabetes, pharmacologic treatment, randomized controlled trials

## Abstract

**Background:** In Type 1 diabetes (T1D), according to the most recent guidelines, the everyday glucose-lowering treatment is still restricted to the use of subcutaneous insulin, while multiple therapeutic options exist for Type 2 diabetes (T2D). **Methods:** For this narrative review we unsystematically screened PubMed and Embase to identify clinical trials which investigated glucose-lowering agents as an adjunct to insulin treatment in people with T1D. Published studies up to March 2022 were included. We discuss the safety and efficacy in modifying cardiovascular risk factors for each drug, the current status of research, and provide a clinical perspective. **Results:** For several adjunct agents, in T1D, the scientific evidence demonstrates improvements in HbA1c, reductions in the risk of hypoglycemia, and achievements of lower insulin requirements, as well as positive effects on cardiovascular risk factors, such as blood lipids, blood pressure, and weight. As the prevalence of obesity, the major driver for double diabetes, is rising, weight and cardiovascular risk factor management is becoming increasingly important in people with T1D. **Conclusions:** Adjunct glucose-lowering agents, intended to be used in T2D, bear the potential to beneficially impact on cardiovascular risk factors when investigated in the T1D population and are suggested to be more extensively considered as potentially disease-modifying drugs in the future and should be investigated for hard cardiovascular endpoints.

## 1. Introduction

The incidence and prevalence of Type 1 diabetes (T1D) is continuously increasing. According to estimates, it affects 15 out of 100,000 persons globally, displaying the highest rates in North America and Scandinavian countries and the lowest in Africa, however, the paucity of data in particular in low- and middle-income countries as well as the considerable number of potentially misclassified types of diabetes might impede reliable estimates [1,2]. In Europe, a doubling of T1D cases has been predicted in children younger than five years of age during the last two decades [3] and noteworthily, the incidence and prevalence of new-onset autoimmune diabetes occurring during adulthood (LADA) is also rising in particular countries [2]. In addition to genetic factors, lifestyle, socioeconomic, and environmental conditions are believed to raise the risk of T1D [4]. While we still do not fully understand the pathophysiology of T1D, major advancements have been made in the management of diabetes over the last few decades facilitating and improving daily diabetes management. The concept of “living with diabetes” has received a significant impetus during this time. The hundred-year anniversary of the first individual receiving insulin has served as a milestone in the history of medicine and in keeping persons alive with a disease that was formerly a death sentence. However, in the early days of insulin treatment, the clinical outcomes were not as favorable as previously assumed: mortality rates were still almost 50% during the first 25 years of disease, and mortality rates secondary to diabetic ketoacidosis (DKA) remained as high as 20–40% in 1940 [5]. Nevertheless, the life expectancy of persons with T1D was prolonged consistently, as demonstrated by the *Pittsburgh Epidemiology of Diabetes Complications* (EDC) study; the duration of life among persons diagnosed with diabetes between 1950 and 1964 was 53.4 years, and rose to 68.8 years among those diagnosed between 1965 and 1980 [6]. More recent data from Sweden indicate a less pronounced but still evident rise in life expectancy for men aged 20 of an additional 47.7 years in 2002–2006 to 49.7 years in 2007–2011 and a non-significant increase in life expectancy for women >20 years of age from 51.7 to 51.9 years [7]. However, there was still an increased mortality evident in people with T1D as shown in a recently published population-based study in England and Wales [8].

The coincidence of T1D and features of metabolic syndrome, referred to as double diabetes, has received significant attention in recent years. In the US, the prevalence of obesity among children with T1D was reported to have increased three-fold (from 12.6 to 36.8%) between the 1980s and the 1990s [9]. The prevalence of overweight and obesity in T1D has developed hand in hand with the growth of the general population [10]. However, registry data from the US and Europe suggest an even higher increase in obesity and overweight in several cohorts of persons with T1D [11,12] as compared to the diabetes-free population. Excess weight gain in children has been suggested as an accelerator of T1D because stress on β-cells and insulin resistance during childhood could potentially increase antigen presentation and trigger disruptions in autoimmunity [13]. In the subsequent course of T1D disease, weight gain is a major issue, especially in persons who follow an intensive treatment aiming to achieve lower HbA_1c_ targets, as seen in the *Diabetes Control and Complications Trial* (DCCT) trial [14]. This is indicative of a vicious cycle: insulin-induced weight gain and subsequent weight-induced insulin resistance require higher doses of insulin to achieve glycemic targets.

Increased insulin resistance or decreased insulin sensitivity in T1D is primarily ascertained by measures of increased body weight together with the amount of insulin needed to achieve and maintain the recommended target glucose levels. While a sedentary lifestyle, nutritional misbehavior, a family history of Type 2 diabetes (T2D), hormonal physiology (puberty, pregnancy), acute disease, and chronic hyperglycemia have been identified as contributors to insulin resistance in T1D, obesity appears to have the strongest impact on the extent of one’s individual insulin resistance. A clear association was observed between insulin resistance and poor glycemic control, as well as the presence of concomitant cardiovascular risk factors such as hypertension and dyslipidemia, also seen in people with T1D [15,16,17]. Furthermore, insulin resistance was shown to be a more powerful predictor of coronary artery disease than glycemic control in persons with T1D [18,19]. Nevertheless, in daily routine, practicable formulae or scores to detail the presence or severity of insulin resistance have not been established yet.

Cardiovascular disease remains the main driver, the prognosis-limiting factor, and the main source of comorbidity and mortality in persons with T1D. Fortunately, the last few decades have witnessed an impressive decline in cardiovascular disease among persons with T1D [20]. This positive trend can be ascribed to multiple factors including novel options to better control glycemia, such as novel insulins, a better understanding of glycemic targets, early detection programs, updated and structured education concepts, greater social perception, and a more careful distribution of co-medication (such as lipid- or blood-pressure-lowering therapy). However, the aforementioned decline in the risk of cardiovascular disease has stagnated in the last two decades and in comparison with the general population, people with T1D still face a considerably elevated risk of cardiovascular disease [20]. Thus, additional attempts are needed to further reduce the morbidity burden in people with T1D.

Several cardiovascular outcome trials (CVOTs) have been performed over the last decade in individuals with T2D. These have demonstrated a tremendous reduction in cardiovascular morbidity and mortality as well as renal events [21,22,23,24,25,26]. Specifically, sodium-glucose co-transporter type 2 (SGLT-2) inhibitors and glucagon-like peptide-1 receptor agonists (GLP-1 RA) exhibited beneficial effects in preventing cardiovascular events in T2D, thus confirming their value as disease-modifying drugs (DMDs) [27]. To date, no such outcome trial has been published for T1D as an adjuvant to standard insulin therapy. Several scientific arguments have been proposed for the positive effects of medical agents typically used in T2D when also applied in T1D. Of note, adjunct agents to insulin in T1D may potentiate improvements in insulin sensitivity, supporting the reduction of insulin requirements, positively influencing weight loss, helping to regulate glucose control, reduce hypoglycemic events, exerting beneficial effects on cardiovascular risk factors, and even improving vascular damage. These effects were observed for several glucose-lowering agents in T1D and deserve further attention.

The present narrative review aimed to highlight the crucial outcomes of randomized controlled trials addressing the role of non-insulin-based glucose-lowering therapies as a component of medical treatment in persons with T1D.

### Non-Insulin Glucose-Lowering Therapies Selected for This Review

This narrative review is confined to a description of agents currently approved and marketed for use in persons with T2D. The selection and sequence of the agents described in the article are based on a review article which intended to list all available pharmaceutical drugs previously introduced and approved for the use in T2D [28]. We have intentionally omitted the description of pramlintide as it is currently not available in most countries worldwide. Medical therapeutics with glucose-lowering properties, but approved for other indications (such as verapamil, bile acid sequestrants, or dopamine agonists) or other agents being evaluated for future use in T2D (such as tirzepatide), as well as agents targeting residual β-cell preservation (immune therapies, verapamil) will not be discussed in this review. Furthermore, non-medical interventions (physical activity, nutrition) or non-pharmacologic agents (such as herbs, vitamins, or nutritional supplements) shall also not be considered here. Figure 1 summarizes the agents used in randomized placebo-controlled clinical trials in people with T1D and Table 1 indicates the number of participants receiving treatment or placebo in these trials. Table 2 refers to specific properties of each drug used in T1D and provides information on the source of evidence. 

## 2. Non-Insulin-Based Glucose-Lowering Agents and Randomized Controlled Trials in Type 1 Diabetes

### 2.1. Sulfonylurea

Drugs belonging to the sulfonylurea (SU) class are characterized by the stimulation of pancreatic β-cells to release endogenous insulin by closing ATP-sensitive potassium channels. Due to this mechanism of action stimulating endogenous insulins secretion, these agents are considered to be unsuitable for the use as add-on therapy in T1D. However extrapancreatic effects, hypothetically increasing the number of cellular insulin receptors, were speculated [86]. *Glibenclamide* has been the only SU which was investigated in T1D and the studies were all of a small sample size [30,31,32,33,34] with the largest trial including 40 participants [33]. While some positive effects on HbA_1c_, in particular in people with preserved insulin secretion, were observed, with a neutral effect on hypoglycemia, research in this field has petered out in the last decades and SUs do not play a role in people with T1D.

### 2.2. Metformin

*Metformin* remains the most widely used glucose-lowering agent in persons with T2D globally. Most T2D treatment guidelines recommend metformin as a first-line pharmacotherapy [87]. The main arguments in favor of this drug are its low cost, positive safety profile, its efficacy in lowering glucose levels, its reported durability, and the positive cardiovascular data mainly derived from the UKPDS study [88]. The glucose-lowering effect of metformin is primarily attributed to a decreased hepatic gluconeogenesis and increased insulin-stimulated peripheral glucose uptake, mainly in the skeletal muscle, by inhibition of the mitochondrial respiratory chain and activation of AMP-activated protein kinase [89].

*Metformin in T1D*: In the context of metformin used in people with T1D, a recent meta-analysis summarizing eight placebo-controlled trials on the application of metformin in T1D provided evidence of significant weight loss (−2.4 kg; (95% CI −4.2 to −0.65)), a total daily insulin reduction (−1.36 IU; (−2.3 to −0.45)), and improvements in LDL cholesterol in the metformin arm. HbA_1c_ and fasting plasma glucose as well as the risk of hypoglycemia and DKA did not differ in the various groups. The metformin groups had a greater likelihood of experiencing adverse gastrointestinal effects and a higher rate of adverse event-related study discontinuations when compared to a placebo [90].

While the majority of the studies included in the above-mentioned meta-analysis were of short duration, a 1-year intervention-based placebo-controlled trial of 100 individuals with poorly controlled T1D (HbA_1c_ ≥ 8.5%) revealed no benefits on glucose control, but similar advantages in terms of reducing body weight and the total daily insulin dose. Notably, severe hypoglycemia accompanied by the loss of consciousness occurred more frequently in the metformin group than in the placebo group (10 versus 2 events; *p* < 0.05). Nearly half of the individuals allocated to the intervention arm reported gastrointestinal side effects, however, not significantly more than in the placebo group [37]. The same study reported improvements in levels of proatherogenic lipid profiles, irrespective of previous background statin therapy [91]. Evidence from a small randomized-controlled study also suggests fewer glycemic fluctuations in metformin users with T1D [92].

A retrospective real-world study addressed the characteristics of metformin users and the effects of metformin on glycemic control during a 10-year follow-up as compared to a population of metformin-naïve patients. At baseline, the metformin group had a higher BMI and required higher insulin doses but had similar HbA_1c_ levels as metformin-naïve patients. Reductions in BMI and insulin doses were noted in the first few years, but the 10-year observation period revealed no sustained effect on HbA_1c_, insulin doses, or BMI [93].

The multicenter, placebo-controlled REMOVAL trial investigated the effect of metformin in a high-risk population of more than 400 persons with T1D, ≥40 years of age, and with three or more cardiovascular risk factors. While intermittent positive effects on HbA_1c_ and insulin dose were observed during the first few months of the study, these parameters did not differ from the placebo group at the end of the study after three years of metformin therapy. However, body weight and LDL cholesterol remained significantly lower three years after the initiation of the study. The progression of mean carotid intima media thickness was not significantly reduced with metformin and endothelial function as measured by reactive hyperemia index was not beneficially modified when compared to the placebo. Concerning microvascular complications, the REMOVAL program showed no difference in the progression of existing retinopathy or the development of new-onset retinopathy. Notably, metformin users experienced a significant increase in their glomerular filtration rates. However, data on albuminuria were not available and any substantial positive effect of metformin on microvascular disease in T1D remains speculative [41]. Notably, in a recently published subgroup analysis of the REMOVAL trial, the progress in cIMT differed according to the smoking status of the participants. Never-smokers who took metformin had significantly slower progress in cIMT than those who used placebo, while previous or current smokers did show a similar extent of cIMT progression as placebo [94]. This finding underlines the detrimental effect of smoking as a risk factor for atherosclerosis in T1D. It might also give rise to an individualized treatment approach: metformin might hinder the progression of atherosclerosis only in non-smokers with T1D. REMOVAL showed gastrointestinal side effects, compelling the premature discontinuation of the trial in 27% of the intervention group (and 12% of the placebo group). Besides, a vitamin B12 deficiency was noted more frequently in persons on metformin therapy (12 vs. 5% over three years), which might be particularly important in persons with T1D, as pernicious anemia, celiac disease, and diabetic gastroparesis are well-known accompanying conditions in T1D. The authors concluded that metformin does not improve glycemic control in long-standing T1D but does appear to play a role in cardiovascular risk management [41].

### 2.3. Thiazolidinediones (TZD)

TZDs or glitazones are selective agonists of the peroxisome proliferator-activated receptor-gamma (PPARϒ), a nuclear receptor that alters the transcription of several genes involved in glucose and lipid metabolism. As an insulin sensitizer, TZDs reduce insulin resistance in adipose tissue, liver, and muscle cells by promoting endocrine signaling from adipocytes [95]. In people with T2D and/or metabolic syndrome, TZDs were shown to improve insulin sensitivity, reduce plasma-free fatty acids and triglycerides, and increase high-density cholesterol (HDL-C). Furthermore, TZDs have been associated with positive effects on endothelial function, inflammation, and the procoagulatory state. Last but not least, TZDs improve both fasting and postprandial hyperglycemia, reduce HbA_1c_, and have a sustainable positive effect on glycemia (good durability) in T2D [96]. TZDs might have beneficial effects in preserving β-cell function in T2D, as suggested by the use of rosiglitazone (DREAM trial) and pioglitazone (ACT NOW trial), and thus slowing down the progression of prediabetes into diabetes [97,98]. While in T2DM some positive effects on cardiovascular endpoints have been attributed to the use of pioglitazone (PROactive study), the incidence of hospitalization for heart failure was increased [99]. Rosiglitazone has already been withdrawn from the market.

To date, only a handful of randomized controlled trials have addressed the effects of TZDs as add-on therapy to insulin in individuals with T1D.

*Pioglitazone in T1D*: In a randomized controlled trial including 60 peripubertal and lean adolescents with T1D who were diagnosed more than one year previously, treatment with 30 mg of pioglitazone once daily for 6 months led to a significant reduction of HbA_1c_ (−0.22 ± 0.29%) and improvements in postprandial plasma glucose in the intervention group. The number of participants who achieved an HbA_1c_ level ≤7% increased from 53 to 70% during pioglitazone treatment while it did not beneficially change in the placebo group. No differences were seen in body weight, hypoglycemic events, insulin requirements, or lipid profiles [50]. In a small, randomized placebo-controlled pilot study comprising newly diagnosed children and adolescents (n = 15) with T1D, the authors explored the effect of once-daily pioglitazone on β-cell preservation by adding pioglitazone to insulin. After 24 weeks of treatment, pioglitazone did not preserve endogenous insulin secretion; insulin requirements and glycemic control were also not improved by pioglitazone therapy [51]. To the best of our knowledge, no trial has addressed the impact of TZDs on vascular protection in populations with T1D.

Another RCT examining 35 adolescents with suboptimally controlled T1D (HbA_1c_ 7.5–11%) and features of insulin resistance (insulin requirements >0.9 IU/kg/d) showed no improvement in glycemic control when pioglitazone was administered for six months and compared to a placebo. In fact, the intervention group experienced an increase in BMI when compared to placebo. Lipid parameters were similar in both groups [49].

*Rosiglitazone in T1D*: Rosiglitazone given twice daily for 8 months in adults with T1D and a BMI ≥ 27 kg/m^2^ resulted in improvements in HbA_1c_. A higher proportion of study participants randomized to rosiglitazone achieved an HbA_1c_ level <6.5% (36 vs. 16%), while the prevalence of hypoglycemia was similar. When compared to baseline levels, weight gain was significantly evident in both groups (rosiglitazone and placebo) [47].

### 2.4. Alpha-Glucosidase Inhibitors (AGIs)

AGIs (acarbose and miglitol) exert their glucose-lowering effect by inhibiting the absorption of carbohydrates from the small intestines, and thus mainly improve postprandial hyperglycemia. AGIs are associated with a modest decrease in body weight, a low risk of hypoglycemia, and positive effects on blood pressure and lipid profiles in people with T2D. While some studies conducted in T2D suggested a risk reduction for myocardial infarction or any cardiovascular event after treatment with acarbose over three years [100,101], the properly powered ACE trial, which was conducted in an Asian population with IGT and coronary artery disease, did not reveal such cardiovascular benefits [102]. AGIs are not included in most of the current guidelines for the treatment of T2D [87].

To date, four randomized placebo-controlled trials (three for acarbose and one for miglitol) have assessed the efficacy and safety of AGIs for use in T1D [29].

*Acarbose in T1D*: The most important placebo-controlled double-blind intervention study, in which acarbose was administered to 114 adults with T1D, showed a significant reduction in HbA_1c_ (−0.48%; placebo-subtracted), as well as fasting and postprandial hyperglycemia, but with no significant effect on insulin dose, risk of hypoglycemia, body weight, or lipid levels during 24 weeks of treatment. Eighty-four percent of the participants randomized to the acarbose group reported side effects, which were mainly confined to the gastrointestinal tract (49% for placebo) [44]. With a number needed to harm of 13, acarbose was the agent with the highest side-effect-related study dropout rates when compared to other glucose-lowering agents (metformin, SGLT2-i, GLP1-RAs, SU, DPP4 inhibitors) used in T1D as shown in a meta-analysis [29].

*Miglitol in T1D*: The only study which investigated miglitol showed a less pronounced concern regarding gastrointestinal side effects, but failed to report any advantages in glucose control, insulin dose, or lipids when compared to placebo [46].

### 2.5. DPP4 Inhibitors

Dipeptidyl-peptidase-4 inhibitors (DPP4-i) inhibit the degradation of the incretin hormones glucagon-like peptide 1 (GLP1) and glucose-dependent insulinotropic peptide (GIP), leading to a stimulation of meal-dependent endogenous insulin secretion, resulting in reduced fasting and postprandial glucose levels. DPP4-i are approved for the treatment of T2D and are characterized by weight neutrality, a low risk of hypoglycemia, and general tolerability. Cardiovascular outcome trials conducted for several DPP4-i have confirmed their cardiovascular safety, but have identified no benefit in terms of reducing major cardiovascular events [103,104,105,106,107]. The use of saxagliptin was associated with a higher risk of hospitalization for heart failure, as shown in the SAVOR-TIMI 53 study [103].

The rationale for using DPP4i in T1D is based on the reduction of postprandial glucagon secretion, exerting a positive impact on glucose control. On the other hand, DPP4i are believed to possess immunomodulatory and other pleiotropic features, potentially improving insulitis and delay β-cell destruction in newly diagnosed persons with T1D who are still able to produce endogenous insulin [108].

*Sitagliptin in T1D*: In a crossover pilot study, eight weeks of treatment with sitagliptin 100 mg daily in 20 adults with T1D reduced postprandial and 24-h glycemia and reduced prandial insulin requirements significantly. The time spent in hypoglycemia was similar in both trial arms [109]. The same research group published a larger and longer placebo-controlled trial (141 participants) which investigated the use of sitagliptin on postprandial glucagon levels and parameters of glycemic control. The area under the curve of post-meal glucagon was comparable for sitagliptin and placebo. HbA_1c_ and insulin doses also did not differ significantly between groups. Additionally, the subgroup of C-peptide-positive participants experienced no beneficial changes in the aforementioned parameters [53].

A small study comprising 18 persons with T1D and preserved insulin secretion investigated the effect of sitagliptin or exenatide plus insulin or insulin plus placebo for one year. Insulin requirements were significantly lower in those treated with sitagliptin or exenatide. However, stimulated C-peptide was not preserved when the participants were given sitagliptin or exenatide [54]. Schopman et al. performed a placebo-controlled study and investigated the effect of sitagliptin on counter-regulatory and incretin hormones in response to hypoglycemic clamp experiments. After six weeks of treatment with sitagliptin, GIP and GLP1 levels were significantly increased in response to hypoglycemia. No such effects were registered for glucagon or adrenergic counter-regulatory hormones [55].

*Vildagliptin in T1D*: An RCT of similar design investigated vildagliptin (2 × 50 mg for 4 weeks) in C-peptide-negative persons with T1D. While glucagon levels in the vildagliptin group were more intensively suppressed during a meal challenge when compared to placebo, glucagon levels in response to a clamp-induced hypoglycemic episode were not significantly altered. Other counter-regulatory hormones were not significantly influenced in response to vildagliptin [52].

*Saxagliptin in T1D*: Treatment with saxagliptin over 12 weeks did not yield benefits in regard of glucose variability, hypoglycemic frequency, or awareness, and also did not improve cognitive function or hormonal counter-regulation in response to hypoglycemia. In addition, treatment with saxagliptin in T1D did not positively influence HbA_1c_, insulin doses, or body weight [56].

### 2.6. GLP1-Receptor Agonists

Glucagon-like peptide-1 receptor agonists (GLP1-RA) exert their main effect by stimulating glucose-dependent insulin secretion from β-cells, and were shown to slacken gastric emptying, inhibit post-meal glucagon release, and reduce food intake [110]. Consequently, GLP1-RAs represent a drug class that is able to significantly reduce body weight. In T2D, CVOTs investigating the GLP1-RAs liraglutide, semaglutide, albiglutide, and dulaglutide have proven efficacy in reducing cardiovascular events [23,24,26,111]. While most of the data presently available were generated in people with obesity and T2D, also some data are available from RCTs performed in T1D.

*Liraglutide*: The ADJUNCT program investigated the effect of liraglutide administered in three doses (0.6, 1.2, and 1.8 mg) in two phase three trials comprising more than 2.000 persons with T1D [62,63]. Both studies included a wide range of persons with T1D, differing in body weight and diabetes duration. The duration of the ADJUNCT ONE trial was 52 weeks, and that of ADJUNCT TWO was 26 weeks. Both studies revealed significant reductions in HbA_1c_, insulin doses, and body weight, but higher rates of hypoglycemia and ketosis (ketone level >1.5 mmol/L), especially in the higher-dosed groups [62,63].

A post hoc analysis of the studies revealed that the efficacy of liraglutide in reducing HbA_1c_ did not depend on baseline body weight and glycemic control subgroups. However, residual β-cell function was associated with a significant impact on HbA_1c_ reduction and the risk of hypoglycemia [69]. Adverse events were higher in the liraglutide groups and were mainly related to the gastrointestinal system. Higher doses of liraglutide (1.2 and 1.8 mg), which were administered over 12 weeks in C-peptide-negative and overweight persons with T1D, were associated with a modest reduction in mean glucose levels and insulin doses, as well as significant weight loss [64].

The NewLira study indicated that liraglutide 1.8 mg, administered for 52 weeks, preserved insulin secretion in adults with newly diagnosed T1D, as shown by sustained stimulated C-peptide secretion and lower insulin requirements when compared to placebo [66]. Liraglutide 1.8 mg had no beneficial effects on blood pressure or IMT when administered over 24 weeks in 100 overweight persons with T1D but did increase heart rate, a common observation with this drug class [112]. In contrast, another study comprising overweight persons with T1D reported the benefits of liraglutide 1.8 mg in terms of reducing systolic blood pressure and markers of obesity, whereas lipids were not influenced by the treatment [113].

A recently published multicenter RCT investigated whether the combination of anti-interleukin (IL)-21 antibody combined with liraglutide would better enable ß-cell survival when compared to IL-21 or liraglutide alone or placebo in recently diagnosed individuals with T1D. After 54 weeks of treatment, the meal-stimulated C-peptide was significantly higher in the combination group but not in the liraglutide or IL-21 group alone when compared to placebo. Body weight was significantly decreased by all active treatment groups. Hypoglycemic events occurred significantly less in the liraglutide arm when compared to placebo [70].

*Exenatide*: The MAG1C RCT investigated the effects of the short-acting GLP1-RA exenatide, administered three times daily over a period of 26 weeks in 108 individuals with T1D who were randomly assigned to receive either exenatide or placebo. While exenatide caused significant reductions in body weight, it had no significant effects on HbA_1c_ or reductions in insulin doses when compared to placebo. Hypoglycemia was observed at the placebo level. No changes in blood pressure or lipids were seen. Study discontinuation rates due to adverse events were higher in the exenatide arm [59,114].

*Albiglutide*: In a study including 67 participants with newly diagnosed T1D, albiglutide administered once a week for one year had no appreciable effect on β-cell function, HbA_1c_ and weight in newly diagnosed individuals with T1D when compared to placebo [71]. Anyway, albiglutide was withdrawn from the market in 2018.

Further GLP1-RAs used in T1D are being investigated in the DIAMOND GLP1 (NCT03668470) study (dulaglutide versus placebo), and the TTT1 (NCT03899402) trial (subcutaneous semaglutide, dapagliflozin, or combination versus placebo). The results of these RCTs are not published yet.

### 2.7. SGLT1/2 Inhibitors

Sodium–glucose linked transporter-2 inhibitors reduce renal tubular glucose reabsorption, leading to glucosuria, and therefore reduce blood glucose in an insulin-independent manner. Further pleiotropic effects include reductions in blood pressure and weight [115]. A large body of evidence from CVOTs in T2D has demonstrated cardiovascular and renal benefits when SGLT2-i were investigated, as well as benefits in people with heart failure [21,25,116,117,118,119,120].

SGLT2-I also showed promising results in persons with T1D. To date, RCT data have been published for nearly all existing SGLT2-i being used in T1D (canagliflozin, dapagliflozin, empagliflozin, ipragliflozin, sotagliflozin).

*Empagliflozin in T1D*: The EASE program, which included three placebo-controlled RCTs (in total 1707 participants), focused on the efficacy and safety of empagliflozin in different doses during 52 weeks of treatment. Improvements in HbA_1c_, time in range, body weight, insulin dose, and systolic blood pressure were seen in all of the dosage arms when compared to placebo. The frequency of severe hypoglycemia was similar in the intervention and placebo arms. However, the rate of DKA was significantly higher with the two higher doses of empagliflozin (10 and 25 mg) when compared to placebo (4.3% with 10 mg, 3.3% with 25 mg, and 1.2% with placebo), while no significant increase was observed with the 2.5 mg dose (0.8%) [74]. Lunder et al. explored arterial function in an RCT comprising 40 participants with T1D; empagliflozin and metformin alone as well as the combination of both were tested against placebo with regards to changes in arterial function. After 12 weeks of treatment, the combination of empagliflozin and metformin improved arterial stiffness to a greater extent than metformin alone and placebo, leading the authors to conclude that empagliflozin potentially provides similar cardiovascular protection, as is a known function of RCTs in patients with T2D [42]. A recently published free-living, placebo-controlled, cross-over study of 8 weeks duration investigating 5 mg of empagliflozin as an adjunct to advanced continuous subcutaneous insulin infusion (CSII) therapy, using either an automated insulin delivery (AID) system or a predictive low glucose suspend (PLGS) system in 39 individuals with T1D, demonstrated a significantly improved time in glycemic range in both trial arms (AID: time in glycemic range was 81% with empagliflozin versus 71% without empagliflozin; PLGS: 80% versus 63%). The risk of hypoglycemia was comparable; DKA occurred in one participant receiving empagliflozin and this was associated with a nonfunctioning insulin pump [76].

*Dapagliflozin in T1D*: The DEPICT studies examined the efficacy and safety of add-on therapy with the SGLT2-i dapagliflozin in persons with poorly controlled T1D over 52 weeks (DEPICT-1 study; 833 participants) and 24 weeks (DEPICT-2; 1465 participants) of treatment. Two different doses of dapagliflozin (5 and 10 mg) improved HbA_1c_, body weight, insulin requirements, and blood pressure. Both doses of dapagliflozin were associated with a higher risk of DKA when compared to placebo [77,121]. A post hoc analysis of the DEPICT study also showed significant reductions in albuminuria [122]. A recently published meta-analysis revealed a higher risk of overall adverse events and serious adverse events in the dapagliflozin-treated participants. Although there was a numerical increase in DKA, the risk ratio did not reach statistical significance, which can be attributed to a limited number of DKA events [123]. *Canagliflozin in T1D*: Rodbard and Henry, who performed research on RCTs by investigating the effects of canagliflozin administered in two doses versus placebo on T1D in trials with a duration of 18 weeks reported similar outcomes in the indices of glycemic efficacy and body weight, as well as reductions in insulin doses and a higher risk of DKA [72,124]. In addition, Rodbard reported improvements in patient satisfaction on the DTSQ questionnaires concerning treatment with canagliflozin [125].

*Ipragliflozin in T1D*: The SGLT2-i ipragliflozin, which is only available in Japan, yielded similar efficacy data and no higher prevalence of DKA. However, the number of participants was rather small in both available studies investigating ipragliflozin [79,126].

*Sotagliflozin in T1D*: Sotagliflozin is a dual inhibitor of SGLT1 and SGLT2. Hence, while SGLT2 is predominantly present in the renal tubules, SGLT1 is involved in intestinal glucose absorption [127]. A multicenter study comprising more than 1400 persons randomly assigned to receive sotagliflozin 400 mg or placebo over 24 weeks showed that the primary endpoint (HbA_1c_ < 7% at week 24 without the occurrence of severe hypoglycemia or DKA) was achieved in more sotagliflozin users compared to placebo users (28.6 vs. 15.2%). In addition, persons receiving sotagliflozin showed significant improvements in body weight, blood pressure, and insulin doses when compared to placebo. Severe hypoglycemia (<55 mg/dL) was less frequent in the sotagliflozin group while the prevalence of DKA was increased with sotagliflozin [81]. The inTandem1 (United States and Canada) and inTandem2 (Europe and Israel) studies investigated the effect of two different doses (200 and 400 mg) of sotagliflozin versus placebo in adults with T1D after a run-in phase of six weeks of insulin therapy optimization. Following 24 weeks of treatment, the percentages of study participants with baseline HbA_1c_ ≥ 7% achieving an HbA_1c_ level <7% were 15.7, 27.2, and 40.3% when placebo or sotagliflozin 200 or 400 mg, respectively, were used (inTandem1). After 52 weeks of treatment, fasting plasma glucose, weight, and insulin dose remained significantly better in the treatment groups. Treatment satisfaction scores were significantly higher in persons taking sotagliflozin. The risk of DKA was higher in persons randomized to sotagliflozin, and the risk of severe hypoglycemia was reduced in the sotagliflozin groups [82,83]. A post hoc analysis of the inTandem program revealed a more favorable efficacy and safety profile when sotagliflozin was used in persons with a BMI > 27 kg/m^2^ compared to those with a BMI < 27 kg/m^2^ [128]. Interestingly, in a recently published small study including 85 participants with not-well-controlled T1D and a younger age (18–30 years) receiving sotagliflozin 400 mg or placebo, the risk of DKA was not increased in the sotagliflozin group. This might be justified by the meanwhile-gained knowledge on DKA risk with sotagliflozin and an improved educational procedure prior to therapy initiation [85].

## 3. Discussion

Due to its autoimmune pathophysiology, T1D is characterized by the destruction of the pancreatic β-cells. While near-normal glucose levels can be hypothetically achieved by the use of novel insulins under control of advanced insulin delivery devices, and modern and accurate glucose-monitoring systems, it appears that glucose control alone is not sufficient to reduce the overall health burden associated with T1D.

The effects of intensive insulin therapy in preventing cardiovascular morbidity and mortality in persons with T1D were clearly proven in the DCCT cohort [129]. Nevertheless, even in persons with T1D and well-controlled glycemia, the risk of death from any cause or cardiovascular cause remains twice as high in persons with well-adjusted diabetes (indicated as HbA_1c_ ≤ 6.9) than in matched diabetes-free controls, as shown in a Swedish observation study [130]. Of note, these data should be viewed with caution. First, the definition of well controlled diabetes does not imply good glycemic control during the entire life of a person with T1D. Secondly, HbA_1c_, the most important parameter of assessing glycemic control, does not fully capture the quality of glycemia as e.g., regular hypoglycemic events tend to decrease HbA_1c_ but might contribute to increase cardiovascular risk. Third and most importantly, the presence of metabolic syndrome does not spare people who suffer from T1D, which is a subject of growing concern as the presence of metabolic syndrome might diminish the positive effects attributable to well-adjusted glycemic control. A recent registry analysis from Australia estimated that as many as 30% of their patients with T1D had features of metabolic syndrome, 89% of them had concurrent arterial hypertension, and 50% suffered from dyslipidemia and obesity. Moreover, with the exception of peripheral neuropathy, all typical diabetes-related complications including micro- and macrovascular complications were more prevalent in persons with T1D who provided features of double diabetes and had concomitant metabolic syndrome [131].

Hence, a “glucocentric” approach restricting to the treatment of glycemia will not suffice and cardiovascular risk factors should be given greater attention in clinical practice. Insulin, which remains the exclusive glucose-lowering agent in persons with T1D, does not permit modification of the metabolic risk profile associated with cardiovascular disease. In contrast, weight issues, which predominantly contribute to further metabolic diseases, are frequently associated with insulin therapy. Therefore, individualized treatment, as suggested in recent guidelines for the treatment of T2D [87], would be most desirable for the treatment of patients with T1D as well. Of note, some agents exclusively intended for use in T2D, have already shown promising results in the use in persons without diabetes: GLP-1 RA also had weight-reducing effects in persons without diabetes. SGLT2-i reduced the risk of heart failure in diabetes-free populations, independent of weight. Metformin and TZDs improved insulin sensitivity in persons who did not fulfill the criteria for T2D. Additionally, as described in this review, several research evidence is available proving efficacy of other agents rather than insulin for the treatment of T1D. However, previously, apart from insulin, the sole glucose-lowering agent approved for the use in T1D (only in the US) was the amylin analog pramlintide [132], which is not approved in most countries worldwide. Finally, in 2019, the European Medical Agency approved sotagliflozin as adjuvant to insulin for T1D in adults, to be used in selected T1D patients with obesity and poorly controlled glycemia. In the same year, dapagliflozin was approved for the use in T1D but after only two years; the marketing authorization holder withdrew the indication of T1D of dapagliflozin due to concerns mainly related to DKA [133].

In general, it can be noted that the use of add on therapy in T1D remains in the background. Data from the “T1D Exchange Clinic Registry” from the US reported a doubling in the number of people with T1D ≥ 26 years of age with a metformin add-on therapy when the periods of 2010–2012 and 2016–2018 were compared, although this was still at a very low level (from two to four percent). Other add-on therapies were used with a frequency of less than one percent [134].

As we know, features of metabolic syndrome in T1D including arterial hypertension, hyperlipidemia, and in particular obesity remain a major challenge, potentially overtopping the risk of glycemia-related detrimental burden on health. Thus, an urgent requirement of future trials investigating add-on therapies in T1D is more detailed scientific scrutiny in the reporting of changes in cardiovascular risk factors in response to the specific glucose-lowering adjunct treatment. A majority of conducted trials did not assess the effects of specific treatments on blood pressure or lipid profiles and only a few studies considered the smoking status of the participants within their baseline characteristics. Most importantly, future research should also prioritize the determination of changes in cardiovascular biomarkers, in order to monitor the progress of vascular damage and to assess cardiovascular endpoints related to a specific drug administered during a clinical trial in people with T1D. To date, hardly any study conducted in T1D examined adjunct treatment effects on vascular/endothelial function and damage. Moreover, no CVOT has been conducted for people with T1D investigating adjunct therapy to standard insulin. Nevertheless, recently published data underlining the benefits of such therapies in terms of beneficially influencing cardiovascular risk factors should not be neglected and it is considered worth to emerge for the next step, striving for a CVOT.

In contrast to the studies which have investigated short-term efficacy on cardiovascular risk factors, such large-scale CVOTs will have to include a larger sample of people with T1D at an increased risk for cardiovascular events. Furthermore, a sufficiently long follow-up duration will need to be chosen in order to achieve the required number of endpoint events.

### Clinical Perspective: What Do the Available Data on Adjunct Therapy in People with T1D Teach Us for Clinical Practice?

Sulfonylurea can be considered as being disqualified for further consideration in the use in T1D by the lack of efficacy shown in the few trials performed. While some small trials showed potential benefits in newly diagnosed people with T1D, which seems understandable if there is remaining insulin secretion, given the pathophysiology of T1D, this drug class can neither contribute to a delay in disease progression nor in the management of cardiovascular risk or in body weight management.

The use of AGIs, specifically acarbose, in T1D has failed to demonstrate positive effects in regards of decreasing weight and insulin dose reduction. The small decrease in HbA1c which was shown in the three available RCTs can be considered as negligible. In addition, with the highest risk of gastrointestinal side-effects among all the agents described in this review, it can be concluded that also AGIs do not play a role in clinical considerations for T1D nowadays.

Using metformin in T1D can contribute to improvements in HbA1c, weight control, and insulin requirements due to its insulin-sensitizing effect. Of note, metformin is one of the agents used in T1D which provides evidence to positively impact on LDL-C; even more, there are some data suggestive of the potential positive cardiovascular effects of metformin when used in T1D, however, this research field needs further attention [135]. Metformin is associated with an increased rate of gastrointestinal side events but can be considered as safe as these side effects are usually transient or reversible after dose reduction or treatment discontinuation. Altogether, metformin appears as a potential treatment option, in particular in obese people with T1D requiring high doses of insulin, although the benefit in this particular patient group should be further investigated.

The use of TZDs as an add-on to insulin in T1D is limited to some small studies which were largely not able to demonstrate positive effects on diabetes-related endpoints or further cardiovascular risk factors. Moreover, the only remaining available TZD, pioglitazone, was solely investigated in adolescents. Hence, TZDs do not represent a treatment option of choice for people with T1D.

Similarly, the group of DPP4-inhibitors had hardly any impact on glucose control when used in T1D and did not prove sufficient efficacy in preserving β-cell function in persons with new-onset diabetes. Some evidence suggests improvements in GLP1 response and a consecutive reduction of postprandial hyperglycemia, however, due to the paucity of evidence and the limited efficacy, we do not consider the use of DPP4-i as game changer in the treatment of T1D.

Specifically, overweight and obese individuals with T1D seem to be appropriate candidates to receive adjunct GLP1-RA therapy, which seem to exert their effects via the inhibition of glucagon secretion or, if still present, also via residual beta-cell function. Both exenatide and liraglutide have shown to positively impact weight control and interestingly, liraglutide was also associated with positive effects on blood pressure in a pooled analysis of a meta-analysis [29]. Gastrointestinal side effects occurring with a similar frequency as in people without diabetes or T2DM have to be taken into account also in the use in T1DM.

Extensive research has been provided for the use of SGLT2 inhibitors as an adjunct to insulin therapy in T1D. The EASE trials for empagliflozin [74], the DEPICT program for dapagliflozin [77], and the inTandem program investigating the double inhibitor treatment of SGLT1 and SGLT2, sotagliflozin [82] have provided evidence for improved glycemic control, reduced insulin doses, and modifying features of the metabolic syndrome (weight and blood pressure). As mentioned before, dapagliflozin and sotagliflozin have been previously approved for the use in T1D and represent important add-on therapies in overweight people with T1D and demands of high insulin doses. As data on this drug class have shown, the beneficial cardiovascular or renal effects are largely independent from the diagnosis of T2D; therefore, it can be speculated that those effects could also be present in people with T1D. A considerable limitation of a broad use of SGLT2-i in T1D remains the increased risk for (euglycemic) DKA. While the majority of RCTs investigating the use of SGLT2-i in T1D revealed higher rates of DKA in the intervention groups, for instance, empagliflozin and dapagliflozin were not significantly associated with a higher DKA risk when data from a meta-analysis were considered [29]. To date, several precipitating factors potentially contributing to the development of SGLT2-i-induced DKA in T1D have been identified, which were a lower BMI, features of insulin resistance, disproportionate insulin reductions, and non-adherence to following sick day rules. Interestingly, HbA1c and the dose of the SGLT2-i per se do not seem to detrimentally impact DKA risk [136]. Thus, to us it appears imperative to further investigate the precipitating factors for DKA in people with T1D using SGLT2-i in order to better understand the risk, rather than terminating this research field entirely. Still, we believe that if used in a selected group of patients, SGLT2-i can represent a valid treatment option in T1D.

## 4. Conclusions

Several glucose-lowering agents previously used for the treatment of T2D have shown beneficial effects in modifying cardiovascular risk factors in T1D as well. Referring to the high and growing prevalence of metabolic syndrome, double diabetes, and cardiovascular comorbidity being present in T1D, these agents should be more extensively considered as being potentially disease-modifying drugs in the future and should be investigated for hard cardiovascular endpoints.

## Figures and Tables

**Figure 1 pharmaceutics-14-01180-f001:**
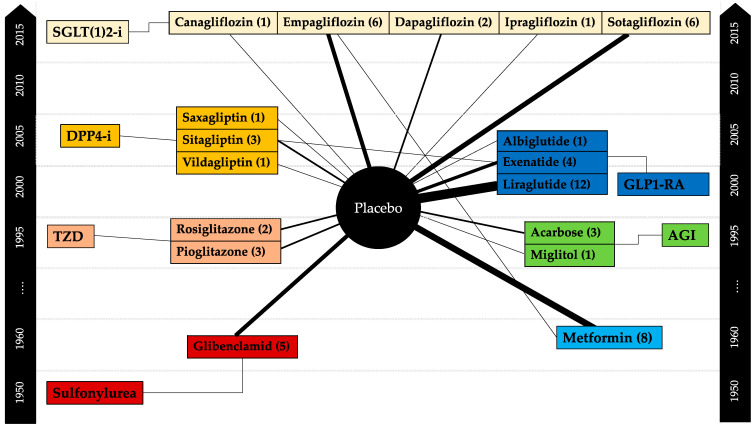
Summary of randomized controlled trials applying adjunct glucose-lowering therapies to insulin in people with T1D. The numbers in braces and the thickness of the lines, which connect the compared agents (or placebo) indicate the number of studies conducted (as modified from Avgernios et al. [29]). The black side bars refer to the chronological sequence of year of market introduction for the use in patients with T2D of the respective substance class (modified from Kahn et al. [28]). AGI = Alpha-glucosidase inhibitors; DPP4-i = Dipeptidyl-peptidase-4 inhibitors; GLP1-RA = Glucagon-like peptide-1 receptor agonists; SGLT2-I = Sodium-glucose linked transporter-2 inhibitors; TZD = Thiazolidinediones (TZD).

**Table 1 pharmaceutics-14-01180-t001:** Summary of the number of randomized placebo-controlled trials and study participants. AGI = Alpha-glucosidase inhibitors; DPP4-i = Dipeptidyl-peptidase-4 inhibitors; GLP1-RA = Glucagon-like peptide-1 receptor agonists; SGLT(1)2-i = Sodium-glucose linked transporter-(1)2 inhibitors; TZD = Thiazolidinediones (TZD).

Agent	Number of RCTs	Total Number of Participants	Number of Participants Receiving Treatment	Number of Participants Receiving Placebo	Years of Publication and References to the Publications
**Sulfonylurea**	5				
Glibenclamide	5	119	65	73	1984 (3×) [30,31,32], 1992 [33], 1995 [34]
**Metformin**	8	719	378	356	2002 [35], 2006 [36], 2008 [37], 2009 [38], 2013(2×) [39,40], 2017 [41], 2018 [42]
**AGI**	4				
Acarbose	3	352	185	195	1991 [43], 1997 [44], 1999 [45]
Miglitol	1	13	13	13	1989 [46]
**TZD**	5				
Rosiglitazone	2	111	55	56	2005 [47], 2015 [48]
Pioglitazone	3	130	66	64	2006 [49], 2007 [50], 2013 [51]
**DPP4-inhbitors**	5				
Vildagliptin	1	28	28	28	2012 [52]
Sitagliptin	3	153	85	84	2013 (2×) [53,54], 2015 [55]
Saxagliptin	1	14	14	14	2016 [56]
**GLP1-RA**	17				
Exenatide	4	156	83	83	2013 [54], 2018 [57], 2019 [58], 2020 [59]
Liraglutide	12	2824	2015	869	2015 (2×) [60,61], 2016 (3×) [62,63,64], 2018 (2×) [65], 2019 (2×) [66,67], 2020 [68], 2021(2×) [69,70]
Albiglutide	1	67	50	17	2020 [71]
**SGLT(1)2- inhibitors**	16				
Canagliflozin	1	351	234	117	2015 [72]
Empagliflozin	6	1861	1328	533	2015 [73], 2018 (4×) [42,74,75], 2022 [76]
Dapagliflozin	2	1336	798	538	2017 [77], 2018 [78]
Ipragliflozin	1	174	115	59	2019 [79]
Sotagliflozin	6	3236	1912	1324	2015 [80], 2017 [81], 2018 (2×) [82,83], 2019 [84], 2021 [85]

**Table 2 pharmaceutics-14-01180-t002:** Efficacy and safety data of non-insulin-based add-on therapies to insulin in the use in people with T1D within randomized controlled trials. The dark colors (green, yellow, red) represent evidence from meta-analyses. The light colors (green, yellow, red) indicate data derived from single studies where only little evidence is available. Blue marked boxes indicate that no evidence is available from meta-analyses or single studies. LDL-C = Low-density-lipoprotein cholesterol. * The three available pioglitazone studies were performed in adolescents.

	HbA_1c_	Total Insulin Dose	Weight	Systolic Blood Pressure	LDL-Cholesterol	DKA Risk	Severe Hypoglycemia	Gastrointestinal side Effects	Treatment Discontinuation due to Adverse Event
**Glibenclamide**									
**Metformin**									
**Miglitol**									
**Acarbose**									
**Rosiglitazone**									
**Pioglitazone ***									
**Liraglutide**									
**Exenatide**									
**Albiglutide**									
**Sitagliptin**									
**Saxagliptin**									
**Vildagliptin**									
**Empagliflozin**									
**Dapagliflozin**									
**Canagliflozin**									
**Sotagliflozin**									
**Ipragliflozin**									
Data derived from meta-analyses		Beneficially changed by intervention		Unchanged by intervention		Adversely changed by intervention		No sufficient evidence available	
Data derived from single studies		Beneficially changed by intervention		Unchanged by intervention		Adversely changed by intervention		No sufficient evidence available	

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
