# Peer review of "Glucose-Lowering Therapy beyond Insulin in Type 1 Diabetes: A Narrative Review on Existing Evidence from Randomized Controlled Trials and Clinical Perspective"

_pharmaceutics, 2022, doi:10.3390/pharmaceutics14061180_

Round 1

Reviewer 1 Report

  • does this review covers only trials up to 2015? as presented in Figure 1? it is unclear;
  • please add the methods for the review, specific keywords, period analysed, etc.;
  • some minor english corrections, such as, line 45, is rise not rose, etc.;
  • do the authors take into consideration also the publications from these trials? 
  • but what about the statistical part of the study? do the trials offers sufficient data in order to be considered as a medical protocol? or used in clinic?
  • I enjoyed reading the review, but since no years/period was mentioned is hard to follow when the studies were made, always having a look at the reference year; I recommend to improve this aspect;
  • I will recommend to make a table as Table 1 but for trials for T2D, in order to point out the first sentence from the abstract, I do not see the point for put it there since it is a review on T1D, but with the first sentence on T2D.

Author Response

does this review covers only trials up to 2015? as presented in Figure 1? it is unclear;

The numbers in the black bars of the y-axis of figure 1 represent the year of marketing of each product for the use in type 2 diabetes (as modified from the given reference). To make this more clearer to the reader we modified following sentence in the caption: “The black side bars refer to the chronological sequence of year of market introduction for the use in type 2 diabetes.” (Table 1)

please add the methods for the review, specific keywords, period analysed, etc.;

Thank you for this important request. This review is designed as narrative and not systematic, however, in order not to miss any relevant publications of RCTs in type 1 diabetes, we used a recent meta-analysis which investigated adjunct-to-insulin therapy in type 1 diabetes (10.1111/dom.14291) which included all randomized controlled studies until the year 2020. After an extensive further literature search starting from 2020 until now, to the best of our knowledge, since then only four further RCTs investigating liraglutide (/10.1016/ S2213-8587(21)00019-X), albiglutide (10.1210/clinem/dgaa149) sotagliflozin (10.1089/dia.2020.0079), and empagliflozin (10.1089/dia.2021.0542) which were incorporated in the article.

some minor english corrections, such as, line 45, is rise not rose, etc.;

Since we are using the past tense in this sentence, we are confident that “rose” should be correct (Line 48). Nevertheless, the manuscript was again read by all contributors in order to check for English language and grammar.

do the authors take into consideration also the publications from these trials? 

Unfortunately, we were not able not understand what is meant by this reviewer comment as no publication was listed for our consideration.

but what about the statistical part of the study? do the trials offers sufficient data in order to be considered as a medical protocol? or used in clinic?

Due to the nature of this narrative review, we neither perform formal statistical analyses nor did we do a meta-analysis. Numerical data was used for Figure 1 and Table 1. Concerning the described studies in the article, there remains a high heterogeneity of sample sizes and statistical methods by the fact that some studies were designed as proof-of-concept trials and some were adequately powered multicentric large-scale trials. The aim of the review was to discuss the various treatment options and the available data published.

I enjoyed reading the review, but since no years/period was mentioned is hard to follow when the studies were made, always having a look at the reference year; I recommend to improve this aspect;

We agree with the reviewer that the time of the specific study publications are hard to determine by reading the manuscript. Therefore, we used table 1 to illustrate the years of study publications and in addition, we provide the respective references related to these trials (Table 1).

I will recommend to make a table as Table 1 but for trials for T2D, in order to point out the first sentence from the abstract, I do not see the point for put it there since it is a review on T1D, but with the first sentence on T2D.

We do agree that this might come across a bit odd and have removed the first sentence on type 2 diabetes. 

Reviewer 2 Report

The paper form Aberer et al. is very interesting and novel, and  surely worthy of publication. The paper is  an in-depth study of the agents currently approved and marketed for  T2D. 

I agree that  insulin, being a potent anabolic hormone, can cause in the long haul obesity, double diabetes and the same  complications as T2D, hence the need to utilize other drugs in combination to possibly mitigate the risks of CVD, to  lower the insulin dose, the HBA1c values, to slow down the onset of micro and macrovascular complications.

This Reviewer is convinced that  our overall lack of knowledge of the etiology of any kind of diabetes  is the ultimate basis of our tendency to act  empirically in this field. 

I suggest to the Authors to render the discussion somewhat less a kind of resume of the results and more a logical examination of how each drug has ameliorated or failed to each of the  risk factors cited in Table 2. Also a  comment on the possible reasons for some  drugs to have been  or not effective, possibly adding an intel on the related mechanisms. This would render the paper much more interesting. Another topic to discusal s a little more  is how the overall hypolglycemic  risk can be considered a major drawback in the combination of drugs with insulin. I have found very inetersting  that in some early T1D cases C-peptide did actually increase. I would like to know whether any study has addressed the C-peptide blood levels before the trial with drugs other  than insulin and has correlated it with the efficacy. For exmaple, of course the use of  secretagogues in principlke would seem not appropriate. 

Author Response

The paper form Aberer et al. is very interesting and novel, and  surely worthy of publication. The paper is  an in-depth study of the agents currently approved and marketed for  T2D. 

I agree that  insulin, being a potent anabolic hormone, can cause in the long haul obesity, double diabetes and the same  complications as T2D, hence the need to utilize other drugs in combination to possibly mitigate the risks of CVD, to  lower the insulin dose, the HBA1c values, to slow down the onset of micro and macrovascular complications.

This Reviewer is convinced that  our overall lack of knowledge of the etiology of any kind of diabetes  is the ultimate basis of our tendency to act  empirically in this field. 

We highly appreciate the positive feedback of this reviewer and agree that this topic has deserved considerable attention specifically in an era of individualized therapy in diabetes.

I suggest to the Authors to render the discussion somewhat less a kind of resume of the results and more a logical examination of how each drug has ameliorated or failed to each of the  risk factors cited in Table 2. Also a  comment on the possible reasons for some  drugs to have been  or not effective, possibly adding an intel on the related mechanisms. This would render the paper much more interesting. Another topic to discusal s a little more  is how the overall hypolglycemic  risk can be considered a major drawback in the combination of drugs with insulin. I have found very inetersting  that in some early T1D cases C-peptide did actually increase. I would like to know whether any study has addressed the C-peptide blood levels before the trial with drugs other  than insulin and has correlated it with the efficacy. For exmaple, of course the use of  secretagogues in principlke would seem not appropriate.

We have revised the discussion according to the suggestions. We do also discuss the hypoglycemia risk together with each drug class. The topic on C-peptide levels and the treatment options to sustain/increase these levels would lead to the field of prevention / delaying of type 1 diabetes, which would be a separate review.  We hope that the reviewer agrees with this procedure.

Reviewer 3 Report

Thank you for the opportunity to review this comprehensive narrative review on adjunctive therapies in T1D. I personally I think it is a highly relevant topic and an underexplored area.  Table 2 very nicely summarises the benefits of different classes of adjunctive agents in T1D.

Some comments for authors to consider

Introduction

  1. The authors report the prevalence of T1D is highest in North America and lowest in Africa and cited systematic review by Mobasseri et al. The reviewer suggest the authors also cite the latest systematic review of incidence of T1D worldwide (Harding et al. Diabetes Care 2022). In this work, the highest rates were in Scandinavian but also unusually high prevalence in certain parts of East Africa. Generally, the lowest incidence was observed in China, Japan and East Asia. The authors should also highlight the difficulties in ascertain true incidence of T1D due to variable definitions and availability of biomarkers

  1. Another aspect that’s worthwhile highlighting is the rising number of T1D individuals diagnosed age >40, often with poorer outcomes. The concomitant presence of other CV risk factors that increase with age and this is an important reason to consider adjunctive therapies. (refer to Harding et al. 2022).

  1. Line 62 “Insulin resistance in T1D is primarily ascertained by clinical investigation and characterized by the administration of supraphysiological high amounts of insulin to achieve and maintain the recommended target glucose level.”

I presume you mean hyperinsulinemic euglycemic clamps here which are only available in research settings. This line may be sound obscure to the non-expert reader. Are there any practical ways that insulin resistance can be assessed in T1D patients in a clinical setting?

  1. Line 72 = “Fortunately, the last few decades have witnessed an impressive decline in cardiovascular disease among persons with T1D1. Improvements mainly attribute to improved insulin kinetics, introucction of diabetes technology, including automated insulin delivery and interstitial glucose monitoring systems.”

Diabetes technology only became more widespread in the last 5 years and I am not aware of robust randomized controlled trial evidence that this was associated with improved long term CV outcomes.  Perhaps improvements in patient education, structured care, use of statins and organ protective agents could have a bigger role. The authors should provide more robust evidence to support their claims.

  1. Line 174 – suggest also highlight the benefits of metformin on weight, which is also an important outcome for people with diabetes.
  2. GLP1-ra lines 271-273 – authors could spend more time elaborating on the GLP1-ra CVOT and renoprotective benefits of GLP1-ra in T2D (eg AWARD-7), given diabetic nephropathy is a major microvascular complication in type 1 diabetes. The authors mentioned one trial investigating effects of GLP1-ra in T1D on IMT. Have there been any other trials that investigated surrogate cardiovascular endpoints?
  3. The authors have nicely summarized trials of adjunctive use of liraglutide, exenatide in T1D. The authors should also summarise any evidence for the newer once weekly GLP1-ra such as dulaglutide and semaglutide. The authors could also mention ongoing trials designed to address these issues ihttps://clinicaltrials.gov/ct2/show/NCT03899402
  4. The authors could also discuss the role of these adjunctive therapies with insulin pumps and other diabetes technologies. Certainly use of these technologies might reduce risk of hypoglycemia, however there is risk of DKA with insulin pump failure. There is at least one trial of adjunctive use of SGLT2i with AID, which showed improved time in range but higher risk of DKA.

Minor

Glibenclamide – typo in Fig 1 and table 1

Formatting lines 424-425

Author Response

Thank you for the opportunity to review this comprehensive narrative review on adjunctive therapies in T1D. I personally I think it is a highly relevant topic and an underexplored area.  Table 2 very nicely summarises the benefits of different classes of adjunctive agents in T1D.

We express our thanks to this reviewer for thoroughly reading and constructively commenting our manuscript and highly appreciate the positive feedback.

Some comments for authors to consider

Introduction

The authors report the prevalence of T1D is highest in North America and lowest in Africa and cited systematic review by Mobasseri et al. The reviewer suggest the authors also cite the latest systematic review of incidence of T1D worldwide (Harding et al. Diabetes Care 2022). In this work, the highest rates were in Scandinavian but also unusually high prevalence in certain parts of East Africa. Generally, the lowest incidence was observed in China, Japan and East Asia. The authors should also highlight the difficulties in ascertain true incidence of T1D due to variable definitions and availability of biomarkers

We thank the reviewer for making us aware about this brand-new and highly-relevant systematic review. We added the high prevalence of T1D in Scandinavian countries and also underlined the inconsistencies concerning the variable number of case reporting and potential failure to assign the appropriate diabetes type (line 33). The recommended reference was included

Another aspect that’s worthwhile highlighting is the rising number of T1D individuals diagnosed age >40, often with poorer outcomes. The concomitant presence of other CV risk factors that increase with age and this is an important reason to consider adjunctive therapies. (refer to Harding et al. 2022).

 We also appreciate this reviewer suggestion to incorporate the circumstance of growing LADA prevalence in our manuscript and we supported this by adding the recommended reference (Line 38).

Line 62 “Insulin resistance in T1D is primarily ascertained by clinical investigation and characterized by the administration of supraphysiological high amounts of insulin to achieve and maintain the recommended target glucose level.”

I presume you mean hyperinsulinemic euglycemic clamps here which are only available in research settings. This line may be sound obscure to the non-expert reader. Are there any practical ways that insulin resistance can be assessed in T1D patients in a clinical setting?

 We thank the reviewer for pointing out this obviously incomprehensible wording. In this matter we did not intend to refer to clamp investigations in order to assess the presence of insulin resistance. In fact, we wanted to state that in clinical practice the co-diagnosis of insulin resistance in type 1 diabetes is mainly defined by high insulin amounts needed to achieve a sufficient glycemic control.  While manifold causes seem to impact on increased insulin resistance or decreased insulin sensitivity in type 1 diabetes, in particular overweight or obesity, practicable formulas or scores to detail the presence or severity of insulin resistance have not been established yet. We have now clarified this in the manuscript (Line 65).

Line 72 = “Fortunately, the last few decades have witnessed an impressive decline in cardiovascular disease among persons with T1D1. Improvements mainly attribute to improved insulin kinetics, introucction of diabetes technology, including automated insulin delivery and interstitial glucose monitoring systems.”

Diabetes technology only became more widespread in the last 5 years and I am not aware of robust randomized controlled trial evidence that this was associated with improved long term CV outcomes.  Perhaps improvements in patient education, structured care, use of statins and organ protective agents could have a bigger role. The authors should provide more robust evidence to support their claims.

 We agree with the reviewer that diabetes technology, due to its short availability might have had no impact on decreasing cardiovascular disease and deleted this part. Furthermore, we expanded the possible explanations. as suggested.

Line 174 – suggest also highlight the benefits of metformin on weight, which is also an important outcome for people with diabetes.

We support the reviewer´s opinion to highlight the beneficial weight effects of metformin when used in T1D. In the previous version of the manuscript, weight loss by using metformin was reported in a meta-analysis (10.1089/dia.2014.0190), an interventional study (10.1371/journal.pone.0003363), an observational study (10.1111/dom.12948) and the REMOVAL studies (10.1016/S2213-8587(17)30194-8). Within the description of all these studies, the positive effects on weight were mentioned (Line 133).  

GLP1-ra lines 271-273 – authors could spend more time elaborating on the GLP1-ra CVOT and renoprotective benefits of GLP1-ra in T2D (eg AWARD-7), given diabetic nephropathy is a major microvascular complication in type 1 diabetes. The authors mentioned one trial investigating effects of GLP1-ra in T1D on IMT. Have there been any other trials that investigated surrogate cardiovascular endpoints?

We agree with the reviewer that the most important property of any antidiabetic drug should be the reduction of micro- and macrovascular disease. Of course, this is valid for people with type 1 and type 2 diabetes. However, to the best of our knowledge, besides the metformin studies, studies on the impact of adjunct therapy to insulin in T1D on cardiovascular surrogate endpoints are scarce / not available. Even more, no such RCT was objecting progression of existing vascular disease or cardiovascular events. This lack of evidence has been mentioned in the discussion (Line 435).  

The authors have nicely summarized trials of adjunctive use of liraglutide, exenatide in T1D. The authors should also summarise any evidence for the newer once weekly GLP1-ra such as dulaglutide and semaglutide. The authors could also mention ongoing trials designed to address these issues ihttps://clinicaltrials.gov/ct2/show/NCT03899402

We thank the reviewer for his proposal to provide an outlook of ongoing trials investigating other GLP1- RAs. We added following section in the manuscript:

Further GLP1-RAs used in T1D are investigated in the DIAMOND GLP1 (NCT03668470) study (dulaglutide versus placebo), and the TTT1 (NCT03899402) trial (semaglutide, dapagliflozin or combination versus placebo). The results of these RCTs are not published yet (Line 312).

The authors could also discuss the role of these adjunctive therapies with insulin pumps and other diabetes technologies. Certainly use of these technologies might reduce risk of hypoglycemia, however there is risk of DKA with insulin pump failure. There is at least one trial of adjunctive use of SGLT2i with AID, which showed improved time in range but higher risk of DKA.

We thank this reviewer so much for his suggestion to include the recently published study examining the use of empagliflozin as adjunct to closed loop systems in individuals with type 1 diabetes (10.1089/dia.2021.0542) in our article. We included following words in the empagliflozin section:

A recently published free-living, placebo-controlled, cross-over study of 8-weeks duration investigating 5 mg of empagliflozin as adjunct to advanced continuous subcutaneous insulin infusion (CSII) therapy using either an automated insulin delivery (AID) system or a predictive low glucose suspend (PLGS) system in 39 individuals with type 1 diabetes demonstrated a significantly improved time in glycemic range in both trial arms (AID: Time in range 81% with empagliflozin versus 71% without empagliflozin; PLGS: 80% versus 63%). Risk of hypoglycemia was comparable; DKA occurred in one participant receiving empagliflozin and was associated with a nonfunctioning insulin pump (Line 336).

Considering this RCT, we have also updated the amount of trials and trial participants in table 1 and figure 1.

Minor

Glibenclamide – typo in Fig 1 and table 1

Thank you for making us aware about this typing error, which we corrected.

Formatting lines 424-425

Thank you, the paragraph was amended.